# Neuromodulation of Astrocytic K^+^ Clearance

**DOI:** 10.3390/ijms22052520

**Published:** 2021-03-03

**Authors:** Alba Bellot-Saez, Rebecca Stevenson, Orsolya Kékesi, Evgeniia Samokhina, Yuval Ben-Abu, John W. Morley, Yossi Buskila

**Affiliations:** 1School of Medicine, Western Sydney University, Campbelltown, NSW 2560, Australia; albabellots@gmail.com (A.B.-S.); 19967644@student.westernsydney.edu.au (R.S.); o.kekesi2@westernsydney.edu.au (O.K.); 19995351@student.westernsydney.edu.au (E.S.); j.morley@westernsydney.edu.au (J.W.M.); 2Projects and Physics Section, Sapir Academic College, D.N. Hof Ashkelon, Sderot 79165, Israel; Uvba.1973@gmail.com; 3International Centre for Neuromorphic Systems, The MARCS Institute, Western Sydney University, Penrith, NSW 2751, Australia

**Keywords:** neuromodulation, astrocytes, potassium homeostasis, spatial buffering

## Abstract

Potassium homeostasis is fundamental for brain function. Therefore, effective removal of excessive K+ from the synaptic cleft during neuronal activity is paramount. Astrocytes play a key role in K+ clearance from the extracellular milieu using various mechanisms, including uptake via Kir channels and the Na^+^-K^+^ ATPase, and spatial buffering through the astrocytic gap-junction coupled network. Recently we showed that alterations in the concentrations of extracellular potassium ([K^+^]_o_) or impairments of the astrocytic clearance mechanism affect the resonance and oscillatory behavior of both the individual and networks of neurons. These results indicate that astrocytes have the potential to modulate neuronal network activity, however, the cellular effectors that may affect the astrocytic K+ clearance process are still unknown. In this study, we have investigated the impact of neuromodulators, which are known to mediate changes in network oscillatory behavior, on the astrocytic clearance process. Our results suggest that while some neuromodulators (5-HT; NA) might affect astrocytic spatial buffering via gap-junctions, others (DA; Histamine) primarily affect the uptake mechanism via Kir channels. These results suggest that neuromodulators can affect network oscillatory activity through parallel activation of both neurons and astrocytes, establishing a synergistic mechanism to maximize the synchronous network activity.

## 1. Introduction

Animal survival is highly dependent on their ability to adapt to the changing environment. To do so, animals are constantly switching between behavioral states, which are correlated to different network oscillations. We recently showed that local alterations in the extracellular K^+^ concentration ([K^+^]_o_) could affect the oscillatory activity of neuronal networks [1] and that specific impairment of the astrocytic clearance mechanism can affect the resonance and oscillatory behavior of neurons both at the single cell and network levels [2], implying that astrocytes can modulate neuronal network activity. However, the cellular and molecular mechanisms that affect this clearance process by astrocytes are still unknown.

Historically, network oscillations were considered to be highly affected by neuromodulation [3], and previous studies indicated an essential role for neuromodulators in mediating the shift between different behavioral states. However, we still know little about the circuitry involved in this neuromodulation, specifically at the cellular level.

Cortical astrocytes found to express a wide variety of receptors for different neuromodulators, including Acetylcholine (Ach, nicotinic α/β and metabotropic M_1–4_) [4,5], Histamine (H_1–3_) [6], Serotonin (5-HT_1,2,5,6,7_) [7], Noradrenaline (NA, α_1,2_-adrenoreceptors and β_1,2_-adrenoreceptors) [8,9], and Dopamine (DA, D_1–5_) [10,11,12], which evoke [Ca^2+^]_i_ increases that affect astrocytic function. For instance, Histamine leads to astrocytic [Ca^2+^]_i_ increases in vitro [13] and mediates the upregulation of the glutamate transporter 1 (GLT-1) through astrocytic H_1_ receptors, leading to reduced extracellular glutamate levels [14] and thus playing a neuroprotective role against excitotoxicity. Similar to Histamine, NA [15,16,17], DA [18,19,20], 5-HT [21,22,23] and ACh [4,24,25] also elicit [Ca^2+^]_i_ elevations in astrocytes, and a recent study showed that superfusion of a cocktail containing these neuromodulators triggered a transient increase in the [K^+^]_o_ levels of mice [26].

Astrocytic Ca^2+^ signaling and glutamate clearance play an essential role in the regulation of the network activity and K^+^ homeostasis, which ultimately affects neuronal excitability underlying network oscillations [1,26,27]. Recently, Ma and colleagues [28] showed that neuromodulators could signal through astrocytes by affecting their Ca^2+^ oscillations to alter neuronal circuitry and consequently behavioral output. In line with these observations, Nedergaard’s group further demonstrated that bath application of neuromodulators to cortical brain slices increased [K^+^]_o_ regardless of synaptic activity [26], suggesting that increased [K^+^]_o_ could serve as a mechanism to maximize the impact of neuromodulators on synchronous activity and recruitment of neurons into networks.

Extracellular K^+^ levels in the brain are mainly regulated by astrocytic activity, including K^+^ uptake via inwardly rectifying K^+^ (K_ir_) channels, Na^+^/K^+^-ATPase (NKA) and Na^+^/K^+^/2Cl^−^ cotransporter (NKCC), and K^+^ spatial buffering to distal areas through gap junctions (GJs), reviewed by [29]. In this study, we have investigated (i) which neuromodulators can affect astrocytic K^+^ clearance mechanisms to amend [K^+^]_o_ levels, including K^+^ uptake through K_ir_4.1 channels that are selectively expressed in astrocytes and responsible for ~45% of K^+^ uptake) [2,29]. Moreover, K^+^ spatial buffering via astrocytic Cx43 GJs, which are the predominant connexins in astrocytes, see [30]; and (ii) does the impact of the different neuromodulators depend on [K^+^]_o_ concentration and thereby different levels of network activity? To this end, we assessed the impact of specific neuromodulators on the astrocytic K^+^ clearance mechanism by measuring their influence on the [K^+^]_o_ clearance time-course in acute brain slices and further assessed whether this impact was due to a direct activation of astrocytic receptors or due to indirect influence via the neural network.

## 2. Results

### 2.1. K^+^ Clearance Time Course in Acute Brain Slices

To measure the [K^+^]_o_ clearance rate, we have used custom-built double-barreled K^+^-selective microelectrodes, as previously described [31]. Transient elevations of [K^+^]_o_ were mediated via local application of KCl at various concentrations corresponding to low (5 mM), high (15 mM) and excessive (30 mM) [K^+^]_o_, which correlated to different levels of network activity. K^+^-selective microelectrodes were placed in layer II/III of the somatosensory cortex, (close to a selected astrocyte stained with SR101 named “astrocyte ∝”, Figure 1A) and were calibrated before and after each experiment, as described in the Materials and Methods section.

Under control artificial cerebrospinal fluid (aCSF) conditions, local application of 30 mM, 15 mM or 5 mM KCl led to [K^+^]_o_ increases of 9.19 ± 0.65 mM (n = 14), 4.38 ± 0.34 mM (n = 16), and 1.38 ± 0.11 mM (n = 15) respectively (*F*(2,42) = 88.15, *p* < 0.0001, one-way ANOVA with Tukey’s post hoc test, Figure 1B,C). The differences between the applied KCl concentrations and the measured [K^+^]_o_ concentrations (by the K^+^ selective electrode located 10 µm away) were due to the dilution of the applied KCl solution as seen in Appendix A.

The [K^+^]_o_ clearance rate was defined as the time it took the [K^+^]_o_ concentration to return to baseline levels measured before the local application of external KCl (90–10% decreasing slope, Figure 1B,D). Under control aCSF conditions, the [K^+^]_o_ clearance rate was concentration-dependent, ranging from 2.02 ± 0.14 mM/s following local application of 30 mM KCl (n = 14) and decreased to 1.09 ± 0.09 mM/s (n = 16) and 0.56 ± 0.05 mM/s (n = 15) following 15 and 5 mM KCl respectively (*F*(2,46) = 62.03, *p* < 0.0001, one-way ANOVA with Tukey’s post hoc test, Figure 1D). These results were consistent with a previous study showing that the decay rate of [K^+^]_o_ was directly correlated to the [K^+^]_o_ amplitude [32].

### 2.2. Alterations in Astrocytic K^+^ Uptake and Buffering Mechanisms Affect the [K^+^]_o_ Clearance Rate

We next assessed the specific impact of astrocytic K^+^ clearance mechanisms, including uptake via Kir4.1 channels and spatial buffering via GJ, on the [K^+^]_o_ clearance rate. Bath application of BaCl_2_ (100 µM) that had been previously shown to selectively block astrocytic K_ir_4.1 channels and thus the uptake into astrocytes [2,33,34,35,36,37,38], significantly (*F*(1,83) = 103.6, *p* < 0.0001 for the factor “Kir4.1 blockade” and (*F*(2,83) = 27.9, *p* < 0.0001 for the factor interaction, two-way ANOVA) reduced the K^+^ clearance rate for all [K^+^]_o_ tested (30 mM, 0.66 ± 0.07 mM/s, n = 18; 15 mM, 0.71 ± 0.04 mM/s, n = 14; 5 mM, 0.28 ± 0.01 mM/s, n = 12; *p* < 0.0001, two-way ANOVA with Tukey’s post hoc test, Figure 2A–C), indicating a slower rate of K^+^ removal from the extracellular space when K^+^ uptake by Kir4.1 channels was impaired.

To decrease K^+^ spatial buffering through the astrocytic network, we incubated the slices with a mixture of connexin-43 (Cx43) mimetic peptides (GAP-26, 200 μM, and GAP-27, 300μM), that inhibited currents carried by connexin-43 hemichannels and GJs [39], thus selectively decreasing astrocytic connectivity via electrical GJs as previously reported [2], see also Figure 2D. Indeed, selective blockade of Cx43 significantly reduced astrocytic connectivity and the K^+^ clearance rate (*F*(1, 82) = 58.54, *p* < 0.0001 for the factor “Cx-43 blockade” and *F*(2, 82) = 28.85, *p* < 0.0001 for the factor interaction, two-way ANOVA). However, disruption of the astrocytic connectivity had a differential impact on the K^+^ clearance rate, as it affected K^+^ transients only at high (15 mM, 0.81 ± 0.04 mM/s, n = 15, Figure 2B) and excessive (30 mM, 0.71 ± 0.08 mM/s, n = 17, Figure 2A) [K^+^]_o_ levels (*p* < 0.0001, two-way ANOVA with Tukey’s post hoc test). Low [K^+^]_o_ concentrations did not lead to a significant change in the K^+^ clearance rate (5 mM, 0.50 ± 0.03 mM/s, n = 11; *p* = 0.84, two-way ANOVA with Tukey’s post hoc test, Figure 2C), confirming the hypothesis that K^+^ uptake via the Na^+^-K^+^ ATPase and Kir4.1 channels was the dominant process used to clear low levels of [K^+^]_o_ and astrocytic K^+^ spatial buffering via GJ takes place at higher levels of network activity [40]. These results were also in agreement with a computational model constructed to simulate the impact of “net uptake” and spatial buffering on [K^+^]_o_ dynamics [41].

### 2.3. Neuromodulation of Astrocytic K^+^ Clearance

#### 2.3.1. Serotonin (5-HT)

Previous studies have demonstrated that astrocytes express several subtypes of serotonergic receptors across different brain areas, including the cortex, corpus callosum, brain stem, spinal cord, and hippocampus, indicating serotonin affects astrocytic activity [7,23,42,43,44,45,46]. To test the impact of 5-HT on the K^+^ clearance rate, we photolyzed (50 µm diameter; UV light) NPEC-caged-Serotonin (30 µM) in layer II/III of the somatosensory cortex, which included the astrocytic domain and nearby neurons (Figure 3A).

Our results showed that following transient application of excessive KCl concentration (30 mM, n = 11) and uncaging of 5-HT, the K^+^ clearance rate decreased to 1.33 ± 0.14 mM/s (*F*(2, 20) = 20.68, *p* = 0.0003, one-way ANOVA with Tukey’s post hoc test, Figure 3B,C). However, co-application of 5-HT and lower KCl concentrations (15 mM and 5 mM) did not affect the K^+^ clearance rate significantly (0.82 ± 0.05 mM/s, n = 11 and 0.34 ± 0.04 mM/s, n = 10, respectively; (*F*(2,20) = 2.549, *p* = 0.1032, one-way ANOVA, Figure 3D,E, Appendix A). Moreover, blockade of neuronal spiking activity with TTX (1 µM) prior to 5-HT photolysis, was comparable with the effect of 5-HT alone, suggesting that the observed alterations in the K^+^ clearance rate at excessive [K^+^]_o_ (~30 mM) were independent of neuronal activity. Further, comparison between the average change in the clearance rate (at excessive [K^+^]_o_) following either blockade of Kir4.1 channels by Ba^2+^, selective blockade of Cx43 channels by a mix of GAP26/27 mimetic peptides, or following photolysis of 5-HT, indicated a significantly higher rate of K^+^ clearance with 5-HT (*F(5, 78)* = 6.453, *p* < 0.0001 one-way ANOVA with Tukey’s post hoc test, Figure 3F). Taken together, these results suggest that the 5-HT impact on K^+^ clearance at excessive concentration was likely due to a direct effect on the astrocytic spatial buffering mechanism, yet the exact pathway in which 5-HT exerts its effect is still unclear.

#### 2.3.2. Dopamine (DA)

DA receptors are classically grouped into D_1_-like (D_1_ and D_5_) and D_2_-like (D_2_, D_3_ and D_4_) receptors that activate opposite signaling cascades [11]. Astrocytes express both types of receptors, as well as Dopamine transporters, but the full impact of Dopamine on astrocytic physiology is still unknown [47].

In order to assess the overall impact of DA on the K^+^ clearance rate we locally uncaged NPEC-caged-Dopamine compounds (10 µM) [48], as described above for 5-HT (Figure 3A). Focal photolysis of caged DA significantly reduced the K^+^ clearance rate at all [K^+^]_o_ concentrations tested, including 30 mM (1.68 ± 0.25 mM/s, n = 13, (*F*(2,24) = 108.5), *p* < 0.0001, one-way ANOVA with Tukey’s post hoc test), 15 mM (1.21 ± 0.15 mM/s, n = 12, (*F*(2,22) = 6.886, *p* = 0.0048, one-way ANOVA with Tukey’s post hoc test) and 5 mM (0.56 ± 0.08 mM/s, n = 14, (*F*(2,26) = 15.15, *p* = 0.0002, one-way ANOVA with Tukey’s post hoc test, Figure 4A–D, Appendix A). Moreover, this effect was independent of neuronal activity, as addition of TTX (1 µM) displayed the same results (Figure 4A–D). Further comparison between the average change in the clearance rate following either blockade of Kir4.1 channels by Ba^2+^ or photolysis of DA indicated a significantly lower rate at excessive concentrations (*F*(4,176) = 10.06, *p* < 0.001, two-way ANOVA with Tukey’s post hoc test, Figure 4E), yet was comparable following the application of either low or high [K^+^]_o_ (Figure 4F,G). Similarly, the average decrease following selective blockade of Cx43 channels by a mix of GAP26/27 was comparable at low and high [K^+^]_o_ (Figure 4F,G), yet significantly lower at excessive [K^+^]_o_ (Figure 4E). These results suggest that DA impact on K^+^ clearance are likely due to a direct effect on astrocytes, which may affect both K^+^ uptake as well as spatial buffering.

#### 2.3.3. Noradrenaline (NA)

Astrocytes express several receptors for NA, including α_1_, α_2_, and β_1_-adenergic receptors, which mediate multiple processes. Recently, the Bekar group showed that application of NA led to an increase of the basal [K^+^]_o_, which was affected by neuronal activity [49].

Our results show that bath application of Noradrenaline bitartrate (40 µM) led to a decrease of the K^+^ clearance rate following local application of excessive (30 mM, 0.80 ± 0.06 mM/s, n = 15, (*F*(2, 28) = 21.12), *p* < 0.0001, one-way ANOVA with Tukey’s post hoc test) or high [K^+^]_o_ (15 mM, 0.70 ± 0.06 mM/s, n = 16, *F*(2, 30) = 23.88, *p* < 0.0001, one-way ANOVA with Tukey’s post hoc test) regardless of neuronal activity, as the addition of TTX did not reverse this effect (Figure 5A–C). However, NA did not affect the K^+^ clearance rate at low [K^+^]_o_ (5 mM, 0.42 ± 0.04 mM/s, n = 15; (*F*(2, 28) = 1.423, *p* = 0.2586, one-way ANOVA with Tukey’s post hoc test, Figure 5D, Appendix A). Further comparison between the average change in the clearance rate following either blockade of Kir4.1 channels by Ba^2+^ or application of NA indicated a significantly lower rate at excessive concentrations (*F(4,176)* = 10.06, *p* < 0.01, two-way ANOVA with Tukey’s post hoc test, Figure 5E), yet was comparable following the application of high [K^+^]_o_ (Figure 5F). Similarly, the average decrease following selective blockade of Cx43 channels by a mix of GAP26/27 was comparable at high [K^+^]_o_ (Figure 5F), yet significantly lower at excessive [K^+^]_o_ (Figure 5E). These results suggest that NA impact on K^+^ clearance was likely due to a direct effect on the astrocytic spatial buffering process via gap junctions activated at high [K^+^]_o_ and to less extent the uptake mechanism through Kir4.1 channels and Na/K ATPase. However, the significant difference between the clearance rate following blockade of GJs and the clearance rate following application of NA at excessive [K^+^]_o_ (Figure 5E) suggest that NA affects GJ connectivity via various pathways and not exclusively via Cx43.

#### 2.3.4. Histamine

Astrocytes express different types of histaminergic receptors, including H_1_, H_2_ and H_3_, which mediate multiple processes, including glutamate clearance [14] and glucose homeostasis [50].

Bath application of Histamine dihydrochloride (50 µM) significantly reduced the K^+^ clearance rate following local application of excessive, high, and low KCl (30 mM, 1.15 ± 0.14 mM/s, n = 10, *F*(2,18) = 12.07, *p* = 0.0005, one-way ANOVA with Tukey’s post hoc test); 15 mM, 0.84 ± 0.08 mM/s, n = 10, *F*(2,18) = 16.60, *p* < 0.0001, one-way ANOVA with Tukey’s post hoc test); 5 mM, 0.30 ± 0.02 mM/s, n = 11, F(2,20) = 17.16, *p* = 0.0001, one-way ANOVA with Tukey’s post hoc test), Figure 6A–D). However, while the histaminergic impact on the K^+^ clearance rate at excessive [K^+^]_o_ was not affected by neuronal activity (30 mM, 1.19 ± 0.16 mM/s, *p* = 0.97, one-way ANOVA with Tukey’s post hoc test, Figure 6B), blockade of neuronal firing with TTX increased the K^+^ clearance rate at high (15 mM, *p* < 0.01, one-way ANOVA with Tukey’s post hoc test, Figure 6C) and low [K^+^]_o_ (5 mM, *p* < 0.01, one-way ANOVA with Tukey’s post hoc test, Figure 6D, Appendix A), indicating the involvement of neuronal activity in mediating the histaminergic effects at these lower concentrations. Further comparison between the average change in the clearance rate following either blockade of Kir4.1 channels by Ba^2+^ or application of Histamine indicated a significant lower rate at excessive concentrations (*F*(4,176) = 10.06, *p* < 0.01, two-way ANOVA with Tukey’s post hoc test, Figure 6E), yet was comparable following the application of low or high [K^+^]_o_ (Figure 6F,G). However, the average decrease following the selective blockade of Cx43 channels by a mix of GAP26/27 was comparable at high and excessive concentrations (Figure 6E,F), yet significantly lower at low concentration (Figure 6G). These results suggest that the histaminergic regulation of astrocytic K^+^ clearance mechanisms is [K^+^]_o_-dependent and involves direct (at excessive concentrations) and indirect (at low and high concentrations) impact via the neural network.

#### 2.3.5. Acetylcholine (ACh)

Astrocytes express both ionotropic receptors (α_,_ β) [4] and muscarinic G protein-coupled receptors (GPCRs) for ACh (M_1–3_) [5,51].

To test the impact of ACh on the K^+^ clearance rate, we bath applied slices with Carbachol (100 µM), a non-specific ACh agonist that binds and activates both nicotinic and muscarinic ACh receptors [52]. However, the K^+^ clearance rate was comparable between normal aCSF and Carbachol conditions for all [K^+^]_o_ tested (30 mM KCl, 1.36 ± 0.13 mM/s; 15 mM KCl, 0.97 ± 0.09 mM/s; 5 mM KCl, 0.51 ± 0.07 mM/s, *F*(2, 28) = 0.6771, *p* = 0.516, One way ANOVA), as shown in Figure 7A–D. Since blockade of neuronal firing with TTX was comparable to control and Carbachol conditions (30 mM KCl, 1.37 ± 0.11 mM/s, n = 15; 15 mM KCl, 1.06 ± 0.10 mM/s, n = 10; 5 mM KCl, 0.54 ± 0.07 mM/s, n = 10; *p* > 0.05, One way ANOVA, Figure 7A–C, Appendix A), these results suggest that ACh has no direct or indirect impact on K^+^ clearance mechanisms.

## 3. Discussion

Neuronal activity is accompanied by a transient local increase in [K^+^]_o_, which must be cleared to maintain neuronal function. In the CNS, K^+^ homeostasis is maintained by astrocytic K^+^ clearance mechanisms, including “net K^+^ uptake” and K^+^ “spatial buffering” to distal areas through GJs [53]. However, the mechanisms that affect these clearance processes and overall [K^+^]_o_ dynamics are largely unknown.

In this study, we investigated the impact of different neuromodulators known to act on both neurons and astrocytes in the K^+^ clearance process. Previous studies have demonstrated that neuromodulators, including DA [54], ACh [55], and NA [56] affect neuronal excitability, leading to altered network oscillations at multiple frequencies [57]. Moreover, modulation of the cholinergic [58,59,60] or monoaminergic [61,62,63] signalling pathways has been reported to affect neural network oscillatory dynamics underlying behavioural shifts, as happens during different phases of sleep (i.e., REM vs. NREM) or between sleep and arousal states. Another key modulator of extracellular K^+^ is the Na^+^/K^+^ ATPase (NKA pump), which is expressed in both neurons and astrocytes, though with different subunit isoforms [64]. However, as there are no selective blockers for the astrocytic Na^+^/K^+^ ATPase, we did not measure its direct effect to avoid misinterpretation of the direct impact of neuromodulators on astrocytic activity.

Recently Ding and colleagues showed that the application of a cocktail of neuromodulators to cortical brain slices result in an increase of [K^+^]_o_, which did not involve neuronal activity [26]. Moreover, different behavioural states, such as arousal and sleep that are modulated by different neuromodulators, were found to be associated with alterations in [K^+^]_o_ dynamics [26,65]. Consequently, we hypothesized that different neuromodulators could modulate [K^+^]_o_ clearance rate by selectively activating different signalling pathways either directly (via astrocytes) or indirectly (via neurons, see [66]) to increase [K^+^]_o_ levels. To validate this hypothesis, we have measured the [K^+^]_o_ clearance rate following local application of KCl at different concentrations in the presence of the neuromodulators 5-HT, DA, NA, Histamine and ACh (Carbachol).

### 3.1. Mechanisms That Affect the K^+^ Clearance Rate

Extracellular K^+^ dynamics are determined by the rate of active K^+^ uptake into nearby astrocytes, as well as the rate of extracellular diffusion [67]. If not cleared, increased extracellular K^+^ can lead to altered network oscillations and ictal activity [66,68]. Previous reports suggested that the rate of K^+^ clearance can also be affected by different factors, including temperature [32], ammonia [69], glutamate [40], and pH, however, the cellular mechanisms affecting this clearance process are still largely unknown.

Among astrocytic K^+^ clearance mechanisms, K^+^ uptake becomes activated following low local increases in [K^+^]_o_ (~3–12 mM), mostly affecting small astrocytic networks located within close proximity to the synaptic release site, and becomes saturated at [K^+^]_o_ above ceiling levels (>12 mM) [70,71]. In contrast, the K^+^ spatial buffering process via GJ-mediated astrocytic networks is active when there is a high accumulation of [K^+^] [29]. In that regard, agents that affect the clearance rate of low [K^+^]_o_ (~5 mM) independent of neuronal activity are likely to play a role in the modulation of astrocytic K^+^ uptake mechanisms, mediated via the NKA pump and Kir4.1 channels [72,73,74]. In contrast, compounds that affect the clearance rate of high and excessive [K^+^]_o_ (15 mM and 30 mM, respectively) are more prone to regulate the K^+^ spatial buffering process through GJs [36,75]. Indeed, our results indicate that selective blockade of Kir4.1 channels affected the [K^+^]_o_ clearance rate at all concentrations tested (Figure 2), as the uptake mechanism occurs at all concentrations. However, selective inhibition of astrocytic GJ’s decrease the clearance rate only at high and excessive [K^+^]_o_ (Figure 2), consistent with previous reports [36,75]. In addition, a key result in this study is that the rate of K^+^ clearance is concentration-dependent and directly correlated to the [K^+^]_o_ (Figure 1). This is probably due the fact that at high concentrations, K^+^ clearance is facilitated by GJ, as previously reported [40,72]. A previous study specified that BaCl_2_ mainly affects the [K^+^]_o_ peak amplitude [72], however, the amplitude was measured following high-frequency stimulation that lasted for 10 s, during which neurons constantly exert K^+^ to the vicinity of the recording electrode. In comparison, our stimulus was much shorter (0.1 s), and, therefore, allows direct measurement of the clearance rate without the impact of further K^+^ increase to the extracellular fluid.

### 3.2. Neuromodulators Impact on the [K^+^]_o_ Clearance Rate

The involvement of different neuromodulators in the regulation of network oscillations has been reported in many studies [49,58,59,60,63], however, the circuitry in which they mediate their impact remains elusive. Here, we demonstrate that certain neuromodulators work in parallel on both neuronal and astrocytic networks, leading to a differential impact on the K^+^ clearance rate. However, while some neuromodulators (e.g., 5-HT, DA, and NA) exert their activity directly via astrocytes, other neuromodulators, such as Histamine expressed a more complex involvement, in which they affect the clearance rate indirectly via neuronal activity at low concentration and directly at an excessive concentration (Figure 6). Moreover, while ACh had no impact on the [K^+^]_o_ clearance rate at any of the concentrations tested (Figure 7), all other neuromodulators significantly decrease the K^+^ clearance rate following the excessive increase of [K^+^]_o_ (Figure 3, Figure 4, Figure 5 and Figure 6). In contrast, the clearance rate following a low increase of [K^+^]_o_ was affected only by DA and Histamine (Figure 4B and Figure 6B, respectively), though DA affected astrocytes directly and the histaminergic effect was mediated by neuronal synaptic activity.

Although our results did not point to a specific pathway by which the different neuromodulators affect the K^+^ clearance (see [12] for a schematic diagram of neuromodulators pathways in astrocytes), our results indicate that at excessive concentrations, none of the neuromodulators tested showed a decrease that was comparable to either blockade of Kir4.1 channels by Ba^+^, or blockade of astrocytic gap junctions by Cx43 mimetic peptides, suggesting that at excessive K^+^ concentrations the impact of the neuromodulators is not necessarily via a complete blockade of the above-mentioned channels and they likely modulate K^+^ clearance via a mixture of pathways that are currently unknown to us, yet affect both the K^+^ uptake and spatial buffering processes. As for low K^+^ concentrations, neuromodulators that showed a clear impact on the clearance rate (e.g., DA and Histamine) had a comparable decrease to the rate measured followed by a selective blockade of Kir4.1 channels (Figure 4G and Figure 6G). Moreover, an addition of Ba^2+^ following incubation with Histamine did not lead to further changes in the clearance rate (data not shown), suggesting their involvement in the modulation of the uptake mechanism. However, we currently cannot exclude the possibility that DA and Histamine affect other pathways as well. At high K^+^ concentrations, all neuromodulators tested (DA, NA, Histamine) had a comparable decrease to the rate measured following incubation with either of the blockers, suggesting that both processes (uptake and buffering) are playing a role at this concentration. However, our data do not indicate a primary pathway through which these neuromodulators affect the clearance rate and more studies with selective blockers to the different pathways are needed to elucidate their role in K^+^ signalling.

Monoamines, including catecholamines (i.e., NA, DA), 5-HT and Histamine are involved in a broad spectrum of physiological functions (e.g., memory, emotion, arousal) [76,77,78], as well as in psychiatric and neurodegenerative disorders (e.g., Parkinson’s disease, Alzheimer’s disease, schizophrenia, depression) [79,80,81]. At the cellular level, the release of neuromodulators impact membrane properties as well as intracellular signalling pathways in both neurons and glial cells [26,28], and previous reports showed that different neuromodulators can fine-tune the hyperpolarization-activated current *I_h_* [82,83,84], thereby affecting membrane resonance of individual neurons, which affect the oscillatory behavior of single neurons and their synchronization into networks [2,12,29,85]. However, whether this was a direct effect of the neuromodulators on neuronal activity, or indirect via astrocytic modulation was never tested.

Our previous study indicated the involvement of astrocytic [K^+^]_o_ clearance in modulating neural oscillations at both cellular and network levels [1,2]. Local transient increase of [K^+^]_o_ affected neuronal excitability and resonance frequency, which facilitated their synchronization and led to high power network oscillations [2]. In that regard, our results in the current study indicate that a reduced clearance rate (specifically at low and high concentrations of K^+^), in the presence of the different neuromodulators (e.g., DA, NA or Histamine) can lead to a temporary increase of [K^+^]_o_, which essentially favours the oscillatory activity of neurons. Consistent with our results, Rasmussen and colleagues showed that elevated [K^+^]_o_ lead to cortical depolarization which was associated with the onset of locomotion and shifts into a desynchronized state which is associated with higher frequency network oscillation [65]. Moreover, Ma and colleagues’ recent study shows that neuromodulatory signalling in Drosophila can flow through astrocytes and promote their synchronous activation [28]. They further suggest that astrocyte-based neuromodulation is an ancient feature of the Metazoan nervous system. Moreover, Slezak and colleagues [86] suggested that astrocytes function as multi-modal integrators, encoding visual signals in conjunction with arousal state. Our results support this concept, in which neuromodulators impact network oscillatory activity through parallel activation of both neurons and astrocytes, in which they alter the dynamics of K^+^ clearance to result in transient increase of [K^+^]_o_, thus establishing a synergetic mechanism to maximize their impact on synchronous network activity and recruitment of neurons into networks.

## 4. Materials and Methods

### 4.1. Ethical Approval

All experiments were approved and performed in accordance with the Western Sydney University committee for animal use and care guidelines (Animal Research Authority #A10588, 15 June 2015). All the experimental procedures complied with *The Journal of Physiology*’s policies on animal research [87].

### 4.2. Animals and Slice Preparation

For this study, we used 4–8-week-old C57/BL6 mice of either sex. All animals were healthy and handled with standard conditions of temperature, humidity, 12 hours light/dark cycle, free access to food and water, and without any intended stress stimuli.

For slice preparations, animals were deeply anesthetized by inhalation of isoflurane (5%), decapitated, and their brains were quickly removed and placed into ice-cold physiological solution (artificial CSF, aCSF) containing (in mM): 125 NaCl, 2.5 KCl, 1 MgCl_2_, 1.25 NaH_2_PO_4_, 2 CaCl_2_, 25 NaHCO_3_, 25 glucose and saturated with carbogen (95% O_2_−5% CO_2_ mixture; pH 7.4). Parasagittal brain slices (300 μm thick) were cut with a vibrating microtome (Leica VT1200S, Mt. Waverly, Australia) and transferred to the Braincubator^TM^ (PaYo Scientific, Sydney, Australia), as reported previously [88,89]. The Braincubator is an incubation system that closely monitors and controls pH, carbogen flow and temperature, as well as irradiating bacteria through a separate UV chamber [90,91]. Slices were initially incubated for 12 min at 35 °C, after which they were allowed to cool to 15–16 °C and kept in the Braincubator^TM^ for at least 30 min before any measurements [92,93].

### 4.3. Electrophysiological Recording and Stimulation

The recording chamber was mounted on an Olympus BX-51 microscope equipped with Infrared/ differential interface contrast (IR/DIC) optics and Polygon 400 patterned illuminator (Mightex, Toronto, Ontario, Canada). Following staining with Sulforhodamine 101 (SR101; Sigma Aldrich, St. Louis, MO, USA) and short recovery period in the Braincubator^TM^, slices of somatosensory cortex were mounted in the recording chamber, for a minimum of 15 min, to allow them to warm up to room temperature (~22 °C) and were constantly perfused at a rate of 2 mL/min with carbogenated aCSF.

[K^+^]_o_ measurements were performed in layer II/III of the somatosensory cortex, by placing the K^+^-selective microelectrode nearby a selected astrocyte (termed “astrocyte ∝”) stained with SR101 (Figure 1A). K^+^ clearance was temperature-dependent, with Q_10_ of 1.7 at 26 °C and 2.6 at 37 °C, mainly due to Na^+^/K^+^ ATPase activity [32]. Due to the absence of a selective blocker for astrocytic Na^+^/K^+^ ATPase, and our interest in assessing K^+^ uptake into astrocytes via Kir channels, these measurements were performed at room temperature (22 °C), when Na^+^/K^+^ ATPase activity was fairly low. Various KCl concentrations, corresponding to low (~5 mM), high (~15 mM), and excessive (~30 mM), were added to physiological aCSF and locally applied at a constant distance (~10 µm away from the K^+^-selective microelectrode and 20 µm above the astrocyte of choice) through a puffing pipette (tip diameter of 1 μm) for 0.1 s, as previously described [2]. Preparation and calibration of the K^+^-selective microelectrodes were performed as detailed in [94,95]. In short, the voltage response of the salinized K^+^-selective microelectrodes was calibrated before and after experiments within the experimental chamber by placing the electrode in aCSF containing different KCl concentrations (2.5 or “normal” aCSF, 4, 10, 15 or 30 mM). Once the electrode potential reached a steady-state, a dose-response curve was calculated using a half-logarithmic (Log_10_) plot. K^+^-selective microelectrodes were considered good if the recorded voltage baseline was stable and the voltage response was similar before and after its experimental usage (~10% deviation). The K^+^ clearance rate was calculated by dividing the concentration amplitude with the decay time (90–10%).

To assess the impact of neuromodulators on the K^+^ clearance rate, [K^+^]_o_ measurements were performed within the same brain slices before and after 5 min bath application of different neuromodulators, including the cholinergic agonist Carbachol (100 μM), Histamine dihydrochloride (50 μM), Noradrenaline bitartrate (40 μM), NPEC-caged-Serotonin (30 μM), and NPEC-caged-Dopamine (10 μM). To exclude the involvement of neuronal activity, similar experiments were conducted after perfusing slices with neuromodulators and tetrodoxin (TTX, 1 μM, Tocris Bioscience, Bristol, UK) for 5 additional minutes. Polygon400 illuminator (Mightex, Toronto, Ontario, Canada) was used to uncage NPEC-caged-Serotonin and NPEC-caged-Dopamine compounds by applying focal photolysis with UV light (360 nm) in a selected area (50 µm), which included the surroundings of the K^+^-selective microelectrode, the KCl puffing pipette, and the selected astrocytic domain with its processes, for 1 s prior to local application of KCl as illustrated in Figure 3A. To assess whether the uncaging light itself affected the clearance process, we measured the clearance rate with focal UV light (360 nm; 50 µm), yet without any caged compound, which indeed indicated an insignificant changes (student t-test) and lack of effect of the uncaging light on the clearance rate.

### 4.4. Drugs

All drugs were stored as frozen stock solutions and were added to aCSF just before recordings. Neuromodulators, including NA, Histamine, 5-HT and DA were purchased from Tocris Bioscience (Bristol, UK). Noradrenaline bitartrate and Histamine dihydrochloride were dissolved in water to a stock solution at final concentration of 100 mM. Carbachol (Sigma Aldrich, St. Louis, MO, USA) and caged neuromodulators, including NPEC-caged-Serotonin and NPEC-caged-Dopamine, were dissolved in DMSO to a stock solution at a final concentrations of 1 M or 100 mM, respectively. All stock solutions were stored at −20 °C and protected from light when required.

### 4.5. Experimental Design and Statistical Analysis

Detailed experimental designs of [K^+^]_o_ measurement are described in Materials and Methods and Results sections, including the number of animals used and cells included in the analysis. These numbers were based on our previous studies and standard practices in the field.

Data were assessed for normality and variance homogeneity using the D’Agostino and Pearson test and Bartlett’s test, respectively. Comparisons of K^+^ clearance before and after application of a certain neuromodulator or TTX were conducted using one-way ANOVA with multiple comparisons, as they were conducted on the same slices and in the same region. For group comparisons between different [K^+^]_o_ concentrations or treatments, in which different slices from different animals were used, we conducted one-way or two-way ANOVA followed by Tukey’s post hoc test, according to the experimental design. Statistical comparisons were made with Prism 7 (GraphPad Software; San Diego, CA, USA), and unless stated, data were reported as mean ± S.E.M. Analysis of K^+^ transient properties were performed using a custom-made MATLAB code (MathWorks). Probability values <0.05 were considered statistically significant.

## 5. Conclusions

Neuromodulators are known to mediate changes in network oscillatory behaviour and thus impact on brain states. In this study we show that certain neuromodulators directly affect distinct stages of astrocytic K^+^ clearance, promoting neuronal excitability and network oscillations through parallel activation of both neurons and astrocytes, thus establishing a synergistic mechanism to maximize the synchronous network activity.

## Figures and Tables

**Figure 1 ijms-22-02520-f001:**
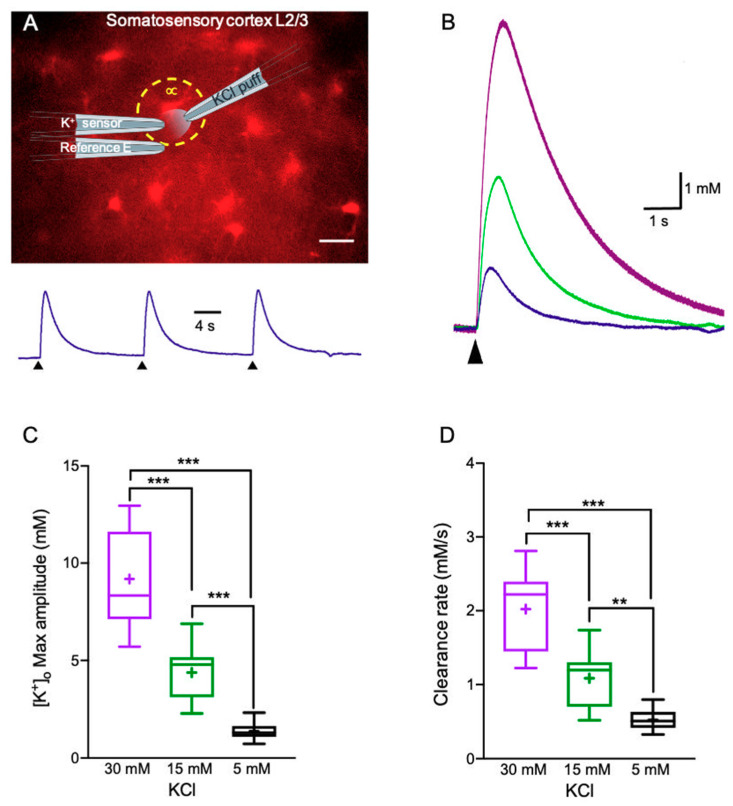
Measuring K^+^ clearance in acute brain slices. (**A**) Top—Fluorescence image of a brain slice depicting astrocytes stained with SR101 along with schematic diagram of the K^+^ recording electrode and the KCl puffing pipette (scale bar—20 μm). Bottom—sample traces of [K^+^]_o_ recordings following repetitive stimulation with 5 mM KCl, arrows indicate the time of KCl application; yellow circle surrounds an astrocyte nearby the recording electrode (astrocyte ∝). (**B**) Average traces of 3 repetitive [K^+^]_o_ recordings depicting the K^+^ clearance time course following local application of 30 mM (purple), 15 mM (green), and 5 mM (blue) KCl puffs. (**C**,**D**) Box plots depicting the maximal amplitude (**C**) and clearance rate (**D**) following local application of 5, 15, and 30 mM KCl. The box upper and lower limits are the 25th and 75th quartiles, respectively. The whiskers depicting the lowest and highest data points, while the + sign represents the mean and the horizontal line through the box is the median. ** *p* < 0.01; *** *p* < 0.001; one-way ANOVA.

**Figure 2 ijms-22-02520-f002:**
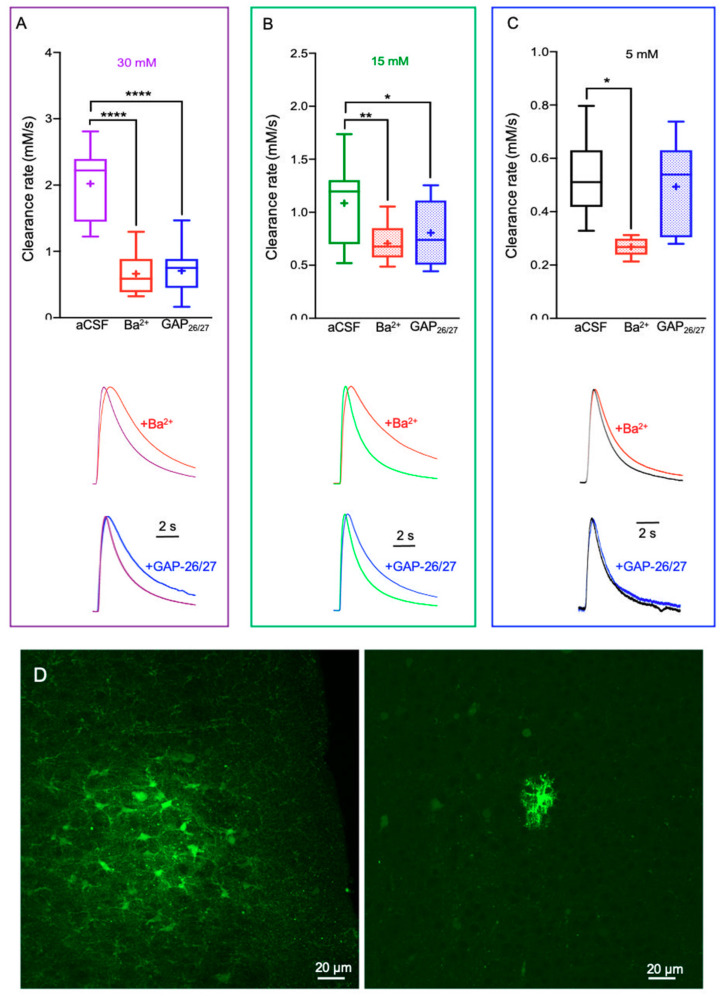
Impaired astrocytic K^+^ clearance affect [K^+^]_o_ dynamics. (**A**–**C**) Box plots depicting the average K^+^ clearance rate following local application of 30 mM (**A**), 15 mM (**B**), and 5 mM (**C**) KCl, under control conditions (aCSF), reduced Kir4.1 activity (red) or impaired astrocytic GJ conductance (blue). Bottom—sample traces of [K^+^]_o_ recording (normalized to the peak amplitude) under control and impaired clearance conditions; red traces- with Ba^2+^; blue traces- with Gap-26/27 mixture. Control traces are color-coded as per their concentration (black—5 mM, green—15 mM and purple—30 mM). Box plots definition are the same as in Figure 1. * *p* < 0.05; ** *p* < 0.01; **** *p* < 0.0001; two-way ANOVA. (**D**) Confocal images (×20) of biocytin-stained astrocytes in the somatosensory cortex depicting the astrocytic network under normal aCSF conditions (left) and following application of GAP 26/27 mixture (right).

**Figure 3 ijms-22-02520-f003:**
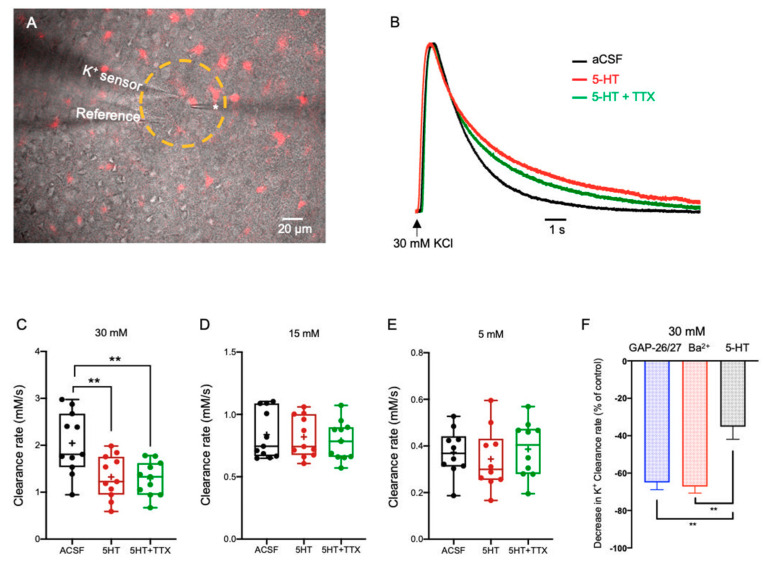
The impact of 5-HT on the K^+^ clearance rate. (**A**) Merged image of DIC (grey) and SR-101 (red) staining depicting the experimental setup, including the K^+^ recording electrode, the KCl puffing electrode (*) and the photo-activated region (yellow circle, 360 nm for 1 s prior to local application of KCl) used for the application of the neuromodulators 5-HT (NPEC-caged-Serotonin, 30 μM) and Dopamine (NPEC-caged-Dopamine, 10 μM). (**B**) Sample traces of [K^+^]_o_ recordings depicting the K^+^ clearance time course following local application of 30 mM KCl (arrow), before (aCSF, black) and after focal photolysis of 30 μM caged Serotonin (5-HT, red) or 30 μM caged 5-HT with 1 μM TTX (green). (**C**–**E**) Box plots depicting the K^+^ clearance rate following local application of 30 mM (**C**), 15 mM (**D**), and 5 mM (**E**) KCl before (black dots) and after photolysis of 5-HT (red dots) or 5-HT + TTX (green dots). (**F**) Bar graph depicting the average decrease in K^+^ clearance rate (as % from control) following blockade of Kir4.1 channels by Ba^2+^(red); blockade of Cx43 channels by a mix of GAP26/27 mimetic peptides (blue), or photolysis of 5-HT (black) together with a local application of 30 mM KCl. Box plot definitions is the same as in Figure 1. ** *p* < 0.01; two-way ANOVA.

**Figure 4 ijms-22-02520-f004:**
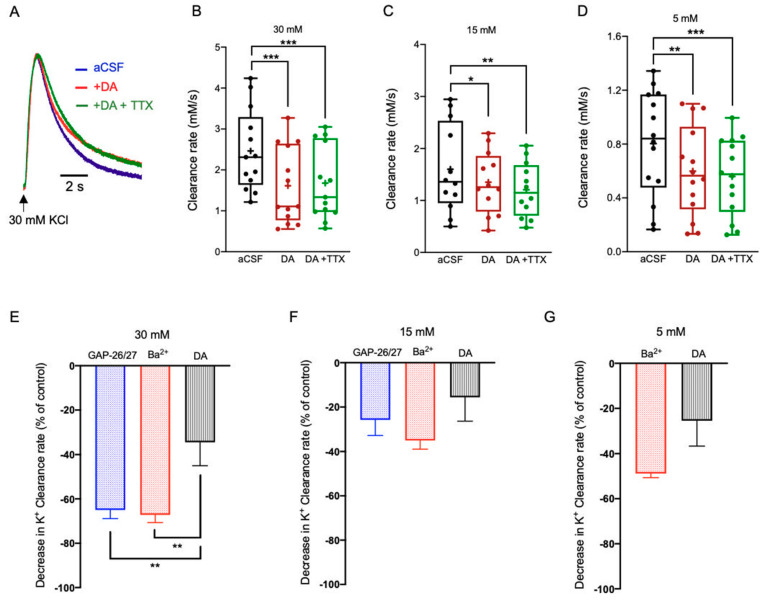
The impact of DA on the K^+^ clearance rate and astrocytic Ca^2+^ signaling. (**A**) Sample traces of [K^+^]_o_ recordings depicting the K^+^ clearance time course following local application of 30 mM KCl (arrow), before (aCSF, black) and after focal photolysis of 30 μM caged Dopamine (DA, red) or 30 μM caged DA with 1 μM TTX (green). (**B**–D) Box plots depicting the K^+^ clearance rate following local application of KCl at 30 mM (**B**), 15 mM (**C**), and 5 mM (**D**), before (aCSF, black dots) and after DA uncaging without (red dots) or with TTX (green dots). (**E**–**G**) Bar graphs depicting the average decrease in K^+^ clearance rate (as % from control) following blockade of Kir4.1 channels by Ba^2+^ (red); blockade of Cx43 channels by a mix of GAP26/27 mimetic peptides (blue), or photolysis of DA (black) together with the local application of 30 mM (**E**), 15 mM (**F**), or 5 mM (**G**) KCl. Box plot definitions is the same as in Figure 1. * *p* < 0.05; ** *p* < 0.01; *** *p* < 0.0001, two-way ANOVA.

**Figure 5 ijms-22-02520-f005:**
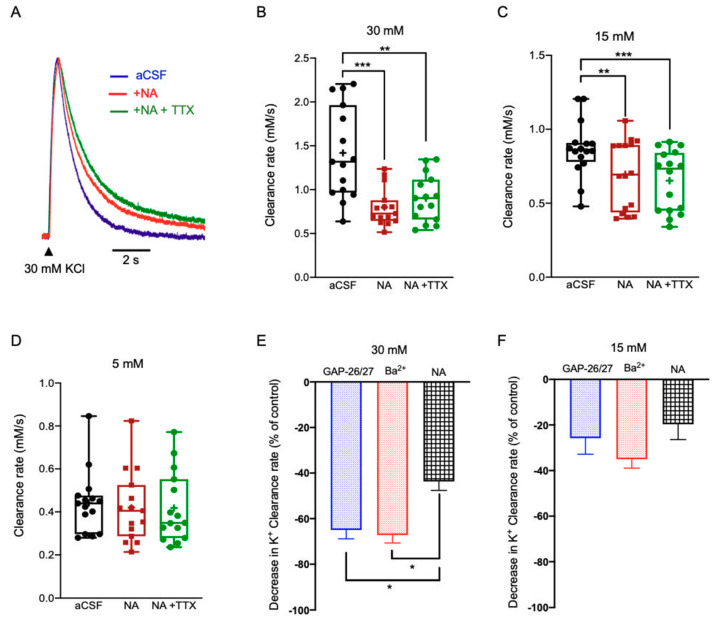
The impact of NA on the K^+^ clearance rate and astrocytic Ca^2+^ signaling. (**A**) Sample traces of [K^+^]_o_ recordings depicting the K^+^ clearance time course following local application of 30 mM KCl (arrow), before (aCSF, blue) and after application of Noradrenaline bitartrate (40 μM, NA, red) or NA with 1 μM TTX (green). (**B**–**D**) Box plots depicting the K^+^ clearance rate following local application of KCl at 30 mM (**B**), 15 mM (**C**) and 5 mM (**D**), before (aCSF, black dots) and after bath application of NA without (red dots) or with TTX (green dots). (**E**,**F**) Bar graphs depicting the average decrease in K^+^ clearance rate (as % from control) following blockade of Kir4.1 channels by Ba^2+^(red); blockade of Cx43 channels by a mix of GAP26/27 mimetic peptides (blue), or photolysis of DA (black) together with the local application of 30 mM (**E**) or 15 mM (**F**) KCl. Box plot definitions is the same as in Figure 1. * *p* < 0.05; ** *p* < 0.01; *** *p* < 0.0001, two-way ANOVA.

**Figure 6 ijms-22-02520-f006:**
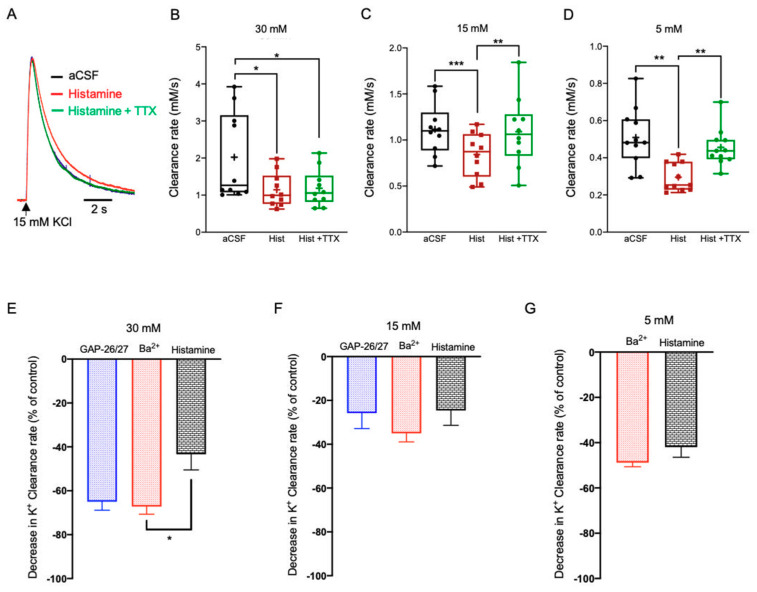
The impact of Histamine on the K^+^ clearance rate and astrocytic Ca^2+^ signaling. (**A**) Sample traces of [K^+^]_o_ recordings depicting the K^+^ clearance time course following local application of 15 mM KCl (arrow), before (aCSF, black) and after bath application of Histamine dihydrochloride (50 μM, red) or Histamine with 1 μM TTX (green). (**B**–**D**) Box plots depicting the K^+^ clearance rate following local application of KCl at 30 mM (**B**), 15 mM (**C**) and 5 mM (**D**), before (aCSF, black dots) and after bath application of Histamine dihydrochloride (50 μM) without (red dots) or with TTX (green dots). (**E**–**G**) Bar graphs depicting the average decrease in K^+^ clearance rate (as % from control) following blockade of Kir4.1 channels by Ba^2+^(red); blockade of Cx43 channels by a mix of GAP26/27 mimetic peptides (blue), or photolysis of DA (black) together with the local application of 30 mM (**E**), 15 mM (**F**) or 5 mM (**G**) KCl. Box plot definitions is the same as in Figure 1. **p* < 0.05; ***p* < 0.01; ****p* < 0.0001; two-way ANOVA.

**Figure 7 ijms-22-02520-f007:**
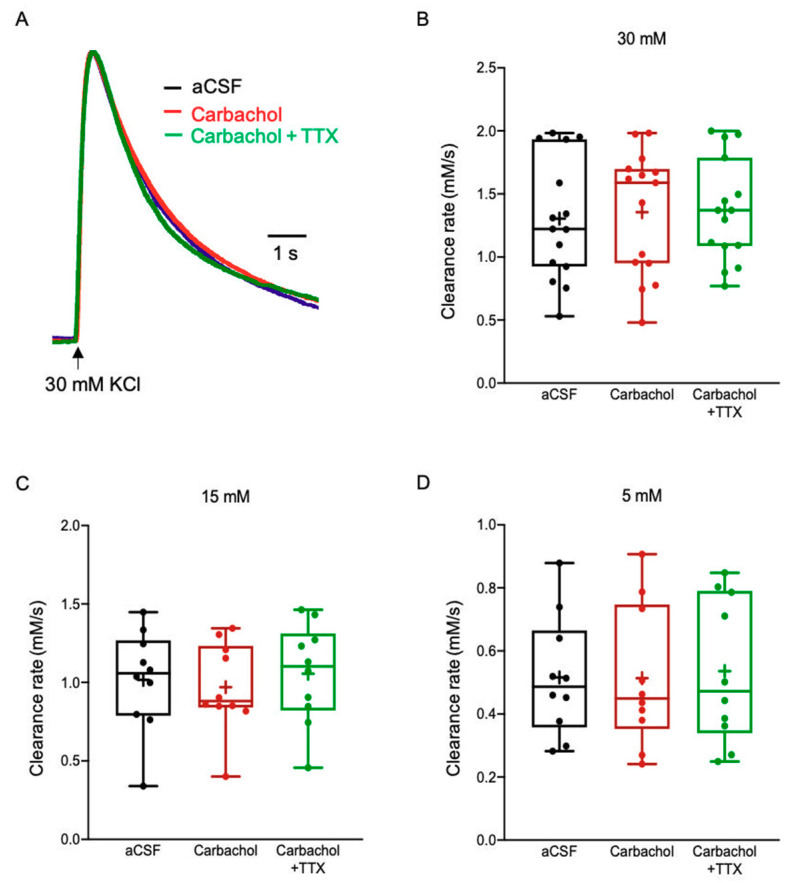
The impact of Carbachol on the K^+^ clearance rate and astrocytic Ca^2+^ signalling. (**A**) Sample traces of [K^+^]_o_ recordings depicting the K^+^ clearance time course following local application of 30 mM KCl (arrow), before (aCSF, blue) and after application of Carbachol (100 μM, red) or Carbachol with 1 μM TTX (green).(**B**–**D**) Box plots depicting the K^+^ clearance rate following local application of KCl at 30 mM (**B**), 15 mM (**C**) and 5 mM (**D**), before (aCSF, black dots) and after bath application of Carbachol (100 μM) without (red dots) or with TTX (green dots). Box plot definitions is the same as in Figure 1.

## Data Availability

Not applicable.

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
