# Peer review of "Neuromodulation of Astrocytic K+ Clearance"

_ijms, 2021, doi:10.3390/ijms22052520_

Round 1

Reviewer 1 Report

Dear Authors

General comments:

 It is very interesting to read about the role of K+ movement regarding interactions between neurons and astrocytes.  In my opinion, more background details have to be given in the Introduction to make it easier to read the manuscript straight through.   This is the more so, as different neuromodulators and their effects on Kir channels as well as Gap Junctions are described.  In the Discussion, please give a scheme (figure or table) illustrating the manuscript's findings.  As said toward the end of the comments, a scheme can be given regarding the pathways and mechanisms modulating K+ clearance, as found in this study, in particular regarding the interactions between neurons and astrocytes in this respect.

Adherence to the general comments would make the paper more accessible and attractive to readers who are interested, but not specialized in the specific subject of K+ movements, In particular in relation to both neurons and astrocytes.

Minor comments, to improve the readability of the paper, are given here below.

The introduction needs a description of Kir channels, including full name and classification , and why the authors chose to work with Kir4.1 channels.

Do the same with gap junctions, why Ba2+ and GAP26 and GAP27 were chosen to modulate GJ activity.  Also mention why Cx43 is targeted.

As different neuromodulators are used, it would be good to present them in a table. 

Regarding reference (26), on page 2, line 50, please say which animals are used.

Line 78, please give a description of aCSF here ( as it appears here first )

Line 116 , please write GJs in stead of gap junctions.

Line 134, it is mentioned that several subtypes of serotonergic receptors are expressed across different brain areas.  Explain shortly why this is relevant for the present study.

Line 233, regarding the word "between"  it should be clearly stated which two items are compared.

Line 403 Please make clear here what the contribution of this study is, and what more needs to be done

In the Discussion, a scheme can be given regarding the pathways and mechanisms modulating K+ clearance, as found in this study, in particular regarding the interactions between neurons and astrocytes in this respect.  This is the more so, as the authors describe different neuromodulators and their effects on Kir channels as well as Gap Junctions.  In my opinion, it is very interesting that various differing pathways can achieve comparable outcomes.  Therefore, I think it is good to see such a pattern in one singular presentation.

Line 447, please give full names of IR/DIC

Line 448, please give full names of (Fluo-4 AM, SR101)

Line 484, the sentence ending with (Figure 3A) please give the reference for this procedure.

Reviewer 2 Report

Neuromodulation of astrocytic K+ clearance by Bellot-Saez et al., show that neuromodulators  such as (5-HT; NA may affect astrocytic spatial buffering via gap-junctions while others like DA; Histamine primarily affect the uptake mechanism via Kir channels. Adding to a growing body of work, they show that neuromodulators affect network oscillatory activity synergistically through parallel activation of both neurons and astrocytes to maximize the synchronous network activity.

The manuscript is well written. The experiments are well executed and controlled and the results are clear. I recommend that the manuscript be accepted for publication.

Author Response

Thank you for your kind evaluation!

Reviewer 3 Report

At the manuscript "Neuromodulation of astrocytic K+ clearance" by Dr. Alba Bellot-Saez et al described investigating of the impact of some neuromodulators which are known to mediate changes in network oscillation on the astrocytic clearance. Obtained results demonstrated that while some neuromodulators (5-HT and NA) might affect astrocytic buffering via gap-junctions, others (Histamine and DA) primarily affect the uptake mechanism using Kir- channels. Authors concluded that neuromodulators can affect neural network oscillation through parallel activation of both neurons and astrocytes.

Impressive work, I have only a few minor comments:

Authors say:

"Monoamines, including catecholamines (i.e. NA, DA), 5-HT and Histamine are involved in a broad spectrum of physiological functions (e.g. memory, emotion, arousal), as well as in psychiatric and neurodegenerative disorders (e.g. Parkinson’s disease, Alzheimer’s disease, schizophrenia, depression). At the cellular level, release of neuromodulators impact membrane properties as well as intracellular signaling pathways in both neurons and glial cells, and previous reports showed that different neuromodulators can fine-tune the hyperpolarization-activated current, thereby affecting membrane resonance of individual neurons, which affect the oscillation of single neurons and their synchronization into networks. However, whether this was a direct effect of the neuromodulators on neuronal activity, or indirect via astrocytic modulation was never tested."

This is not entirely true, since studies of the possible role of neuromodulators on the oscillation of neurons were carried out in an indirect way. Including there were studies of pathological oscillations in epilepsy and the role of extracellular potassium. I would suggest using article (Volnova et al, Brain Sci. 2020 Dec 7;10(12):942. doi: 10.3390/brainsci10120942) for discussion.

Also, I suggest to use in the discussion the new data on the role of extracellular potassium published in: (Huiming Li et al, Neuropharmacology. 2019 Jun; 151: 144-158. doi: 10.1016 / j.neuropharm.2019.04.017. Epub 2019 Apr 15.).

Minor criticisms:

in all figures there is no voltage calibration, maybe it is better to add?

it is not entirely clear what the scale bar in Figure 1B means. There should be voltage along the vertical, but it says 1mM?

in Fig. 5 A the letters in the designation aCSF (blue color) have shifted, probably need to be corrected

Besides that the presentation of a subject is systematic and comprehensive, list of references is quite full and statistical analysis is proper. I am happy to recommend the manuscript for the publication after corrections mentioned above.
